# CRISPR-Based Multi-Gene Integration Strategies to Create *Saccharomyces cerevisiae* Strains for Consolidated Bioprocessing

Odwa Jacob, Gert Rutger van Lill  and Riaan den Haan *

Department of Biotechnology, University of the Western Cape, Bellville, Cape Town 7530, South Africa
* Correspondence: rdenhaan@uwc.ac.za; Tel.: +27-21-959-2199

**Abstract:** Significant engineering of *Saccharomyces cerevisiae* is required to enable consolidated bioprocessing (CBP) of lignocellulose to ethanol. Genome modification in *S. cerevisiae* has been successful partly due to its efficient homology-directed DNA repair machinery, and CRISPR technology has made multi-gene editing even more accessible. Here, we tested the integration of cellulase encoding genes to various sites on the yeast genome to inform the best strategy for creating cellulolytic strains for CBP. We targeted endoglucanase (EG) or cellobiohydrolase (CBH) encoding genes to discreet chromosomal sites for single-copy integration or to the repeated delta sites for multi-copy integration. CBH1 activity was significantly higher when the gene was targeted to the delta sequences compared to single gene integration loci. EG production was comparable, though lower when the gene was targeted to a chromosome 10 site. We subsequently used the information to construct a strain containing three cellulase encoding genes. While individual cellulase activities could be assayed and cellulose conversion demonstrated, it was shown that targeting specific genes to specific loci had dramatic effects on strain efficiency. Since marker-containing plasmids could be cured from these strains, additional genetic changes can subsequently be made to optimize strains for CBP conversion of lignocellulose.

**Keywords:** consolidated bioprocessing; heterologous cellulase production; multi-gene expression; CRISPR-Cas9; *Saccharomyces cerevisiae*

## 1. Introduction

The industrial organism *Saccharomyces cerevisiae* has traditionally been used in wine-making, baking, and brewing [1,2]. This yeast has also been selected as a host for the development of first- and second-generation biofuel production, which utilizes either food biomass (corn starch, sugarcane) or lignocellulosic feedstocks (straw, corn stover, wood), respectively. *S. cerevisiae* is also used to produce enzymes and pharmaceutical proteins such as hepatitis vaccine components, and it is being explored to make advanced biofuels, including farnesene and isobutanol, as well as fine chemical compounds such as resveratrol or nootkatone [2]. The production of fuels and chemicals in biorefineries necessitates the use of robust strains that are resistant to industrial stresses, such as low pH, high ethanol concentrations, fluctuating temperatures, and the presence of different inhibitors [3]. To create strains for biorefinery applications, several genetic manipulations may thus be required to enable pathway alterations that will optimize the production of the desired product.

Bioconversion of lignocellulosic biomass is gaining traction as a technique for generating biofuels and higher-value commodity products [4]. Biomass processing to produce bioethanol includes (i) chemical and/or physicochemical pre-treatment of biomass to make it susceptible to cellulolytic enzyme activity; (ii) enzymatic hydrolysis of pre-treated biomass components; and (iii) fermentation of the resulting hexose and pentose sugars [5]. The most basic conversion scheme is separate hydrolysis and fermentation (SHF), where each process is performed independently, allowing optimal conditions for the individual processes. However, it poses difficulties for industrial applications that include low

conversion rates and the risk of contamination. By combining enzymatic hydrolysis with microbial fermentation of hexoses, simultaneous saccharification and fermentation (SSF) streamline and shorten the process stages [6]. However, both SHF and SSF require expensive exogenous enzymes, indicating that there is room for improvement in capital and operating costs. The most integrated bioconversion process is consolidated bioprocessing (CBP). All CBP processes, including enzyme synthesis, occur in a single reactor. CBP conversion of pre-treated lignocellulose using a single microorganism or microbial consortium capable of producing the required enzymes and fermenting the resulting sugars into value-added products may provide economic benefits. However, CBP requires a microbial workhorse with the necessary phenotypes for enzyme synthesis, saccharification, and productivity [7–10]. The major challenges in developing the industrial ethanologen *S. cerevisiae* for effective microbial conversion of lignocellulosic biomass include heterologous expression of cellulolytic enzymes, engineering co-fermentation of hexose and pentose sugars, and ensuring resilience to different stressors [10]. Cellulose, hemicellulose, and lignin are the three major components of lignocellulosic biomass. Lignocellulose is composed mostly of cellulose, which is a β-1,4 linked polymer of glucose, and glucose is the most abundant sugar in lignocellulose hydrolysates [6]. The release of glucose monomers from cellulose requires the synergistic and coordinated activity of three main types of cellulases, namely: endoglucanases (EGs), exoglucanases such as cellobiohydrolases (CBHs), and β-glucosidases (BGLs) [9]. Heterologous expression of genes encoding cellulases from fungi or bacteria has been shown in recombinant strains of *S. cerevisiae*, including combined expression to create rudimentary cellulase systems [8,10]. Strategies of heterologous cellulase expression, as well as examples of conversion of cellulosic substrates to ethanol by recombinant yeast strains, have been reviewed [11,12]. Stated concisely, strains with rudimentary cellulase systems could partially digest cellulosic feedstocks, but the complete conversion of crystalline substrates without the addition of exogenous cellulases remains elusive. Therefore, additional advancements and rapid utilization of lignocellulosic sugars by fermenting microorganisms in industrially relevant conditions are needed before adopting engineered yeasts for commercially viable bioconversion on an industrial scale. However, the scope for additional genetic manipulations in these strains is often limited because of the use of traditional vectors that curb further engineering due to a lack of available markers.

To direct the metabolic flux towards products of interest, strain improvement through metabolic engineering may require multiple rounds of genetic changes, such as (i) the introduction of heterologous genes or whole metabolic pathways, (ii) the complete or partial elimination of the activity of some endogenous proteins and/or (iii) the overproduction of certain endogenous proteins [13]. To enable CBP conversion of lignocellulosic sugars to products of interest, many genetic changes must be engineered into this yeast. While several methods to enable genetic engineering in yeasts have been developed over the past four decades, multiple changes often require multiple rounds of engineering that become cumbersome or even impossible due to a lack of appropriate vectors or selection markers. The genetic malleability of *S. cerevisiae* is greatly aided by a preference for homologous recombination (HR) over non-homologous end joining (NHEJ) for double-stranded break (DSB) repair [14]. Researchers have taken advantage of *S. cerevisiae's* preference for HR, as it allowed for site-specific integration of foreign genetic material into the yeast genome to produce desired recombinant yeast strains.

The development of the CRISPR-Cas9 system for genome editing through the introduction of targeted DSBs and subsequent break repair has resulted in the publication of numerous studies showing a broad range of applications, including gene knockouts as well as knock-ins [15–17]. Several methods for editing *S. cerevisiae* using CRISPR-Cas9 systems have been established. The bulk of them utilize different constructs to express the gRNA and Cas9 endonuclease. The development of multi-copy integration of heterologous genes in the yeast *S. cerevisiae* using the CRISPR-Cas9 genome editing tool has been revolutionary. It has facilitated the development and advancement of yeast cell factories capable of con-

verting a wide range of substrates into a wide range of products ranging from fuels and chemicals to drugs [3].

In this study, we tested the effect of integrating cellulase encoding genes in various genomic loci in single or multi-copy, in lab and industrial strain backgrounds. We observed variation in the activities obtained when genes were integrated into different loci and that this changed for different reporter genes. Using this knowledge, we created strains with a rudimentary cellulase system that were able to hydrolyze crystalline cellulose. Due to the marker-free nature of CRISPR-Cas9 engineering, subsequent manipulation of these strains is also possible.

## 2. Materials and Methods

### 2.1. Plasmids, Microbial Strains, and Primers Used

Tables 1 and 2 summarize the origins and details of the plasmids and *S. cerevisiae* strains used in this study. All primers used in the study, including their names, sequences, annealing temperatures, and applications, are described in Supplementary Table S1.

**Table 1.** Plasmids used in this study.

| Plasmid | Description | Reference |
|---|---|---|
| pRDH180 | *eg2* plasmid, carrying the *ENO1* promoter, terminator and *T.r.eg2*. Used to produce PCR products carrying the *eg2* gene cassette. | [18] |
| pMI529 | *cbh1* plasmid, carrying the *ENO1* promoter, terminator, and *T.e.cbh1*. Used to produce the PCR product carrying the *cbh1* gene cassette. | [19] |
| pIBG-SSAD | *bgl1* plasmid, carrying the *SED1* promoter, *DIT1* terminator, and *A.a.bgl1*. Used to produce the PCR product carrying the *bgl1* gene cassette. | [20] |
| pCas9NAT | CEN6/ARS4 plasmid, *TEF1* promoter, *CYC1* terminator, SV40 Nuclear Localization Sequence, human codon optimized *S.p.Cas9*; CloNAT resistance. Low copy plasmid carrying the *cas9* encoding gene for the 2-plasmid CRISPR system. | Addgene |
| pRS42-G_ChX | Guide RNA expression plasmid for the 2-plasmid system targeting Chromosome X (Ch10) intergenic region; G418 resistance; contains the *SNR52* promoter and *SUP4* terminator for gRNA expression | [21] |
| pRS42-G-DELTA | Similar to pRS42G_ChX, but targeting the yeast DELTA sequences | This study |
| pRSCG_ChXI | pRS423-cas9-gRNA-G418 targeting Chromosome XI (Ch11) intergenic region protospacer; G418 resistance; this plasmid also contains the *S.p.Cas9* under *TEF1* promoter and *CYC1* terminator – 1-plasmid system. | [21] |
| pRSCG_ChXII | Similar to pRSCG_ChXI but targets a Chromosome XII (Ch12) intergenic region protospacer. | This study |

## 2.2. Microbial Cultivation

All plasmids (Table 1) were propagated using *Escherichia coli* DH5α (Thermo-Fischer Scientific, Waltham, MA, USA) and were cultured overnight at 37 °C on LB agar (5 g/L yeast extract, 10 g/L tryptone, 10 g/L sodium chloride, and 20 g/L agar) containing 100 µg/mL ampicillin. The bacterial colonies from overnight incubation were inoculated in TB media (24 g/L yeast extract, 12 g/L tryptone, and 4% (*v/v*) glycerol) containing 100 µg/mL ampicillin and incubated at 37 °C on a rotary wheel overnight, prior to plasmid DNA extraction. The yeast strains listed in Table 2 were obtained from 15% (*v/v*) glycerol stocks stored at −80 °C and cultivated on YPD (10 g/L yeast extract, 20 g/L glucose, 20 g/L peptone, and 20 g/L agar when required) supplemented with 100 µg/mL CloNAT (Jena Bioscience, Jena, Germany) and/or 200µg/mL Geneticin (G418) disulphate (Invitrogen, Waltham, MA, USA) as needed, at 30 °C for 2–3 days.

## 2.3. Plasmid Preparation and PCR Amplification of Repair Templates

All plasmids were extracted using the ZymoPure Plasmid Maxiprep kit (Zymo Research, Irvine, CA, USA) as directed by the manufacturer. For all PCR analyses performed, Taq DNA Polymerase Master Mix RED (Ampliqon, Odense, Denmark) was used according to the manufacturer's instructions in an Applied Biosystems (Waltham, MA, USA) thermocycler. The plasmids pRDH180, pMI529 and pIBG-SSAD, were used to amplify the homology repair templates for *Trichoderma reesei eg2* (*T.r.eg2* GenBank:KX255673), *Talaromyces emersonii* (now called *Rasamsonia emersonii*) cbh1 (*T.e.cbh1* GenBank:AAL89553), and *Aspergillus aculeatus BGL1* (*A.a.bgl1* GenBank:D64088). Each of these plasmids was assigned specific primers for amplification of their target gene cassette (see Supplementary Table S1). The PCR reactions for the templates pRDH180 and pMI529 were conducted as follows: an initial denaturation step at 95 °C for 5 min, followed by 31 cycles of denaturation at 95 °C for 30 s, annealing at 58 °C for 30 s, and extension at 72 °C for 2 min 45 s; a final extension step of 7 min at 72 °C was allowed. For pIBG-SSAD as a template, the conditions were optimized and set as follows: an initial denaturation step at 94 °C for 5 min, followed by 31 cycles of denaturation at 95 °C for the 30 s, annealing at 55 °C for 30 s, and extension at 72 °C for 3 min 10 s; a final extension step of 7 min at 72 °C was allowed. PCR products were resolved by gel electrophoresis to confirm the amplification. The PCR products were then purified using standard Phenol: Chloroform: Isoamyl alcohol (PCI) extraction, followed by quantification on a Nanodrop2000 Spectrophotometer (Thermo-Fischer Scientific) to determine the concentration of the purified PCR product to be used for transformation.

**Table 2.** Yeast strains used in this study.

| Strain | Abbreviation | Description | Reference |
|---|---|---|---|
| *S. cerevisiae* MH1000 | MH1000 | Industrial yeast strain, diploid, no auxotrophy | [22] |
| *S. cerevisiae* M1744 | M1744 | Haploid yeast strain with uracil auxotrophy (Δ*ura3*) | [18] |
| *S. cerevisiae* M1744 + pCas9 + pRS42-G_ChX + *T.r.eg2* | M1744-Ch10-EG2 | *S. cerevisiae* M1744 with the *T.r.eg2* integrated at the chromosome X intergenic site using pCas9NAT and pRS42H_ChX | This study |
| *S. cerevisiae* M1744 + pRSCG_ChXI + *T.r.eg2* | M1744-Ch11-EG2 | *S. cerevisiae* M1744 with the *T.r.eg2* integrated at the chromosome XI intergenic site using pRSCG_ChXI | This study |

**Table 2.** *Cont.*

| Strain | Abbreviation | Description | Reference |
|---|---|---|---|
| *S. cerevisiae* M1744 + pRSCG_ChXII + *T.r.eg2* | M1744-Ch12-EG2 | *S. cerevisiae* M1744 with the *T.r.eg2* integrated at the chromosome XII intergenic site using pRSCG_ChXII | This study |
| *S. cerevisiae* MH1000 + pCas9 + pRS42-G_ChX + *T.r.eg2* | MH1000-Ch10-EG2 | *S. cerevisiae* MH1000 with the *T.r.eg2* integrated at the chromosome X intergenic site using pCas9NAT and pRS42H_ChX | This study |
| *S. cerevisiae* MH1000 + pRSCG_ChXI + *T.r.eg2* | MH1000-Ch11-EG2 | *S. cerevisiae* MH1000 with the *T.r.eg2* integrated at the chromosome XI intergenic site using pRSCG_ChXI | This study |
| *S. cerevisiae* MH1000 + pRSCG_ChXII + *T.r.eg2* | MH1000-Ch12-EG2 | *S. cerevisiae* MH1000 with the *T.r.eg2* integrated at the chromosome XII intergenic site using pRSCG_ChXII | This study |
| *S. cerevisiae* M1744 + pCas9 + pRS42-G_ ChX + *T.e.cbh1* | M1744-Ch10-CBH1 | *S. cerevisiae* M1744 with the *T.e.cbh1* integrated at the chromosome X intergenic site using pCas9NAT and pRS42H_ChX | This study |
| *S. cerevisiae* M1744 + pRSCG_ChXI + *T.e.cbh1* | M1744-Ch11-CBH1 | *S. cerevisiae* M1744 with the *T.e.cbh1* integrated at the chromosome XI intergenic site using pRSCG_ChXI | This study |
| *S. cerevisiae* M1744 + pRSCG_ChXII + *T.e.cbh1* | M1744-Ch12-CBH1 | *S. cerevisiae* M1744 with the *T.e.cbh1* integrated at the chromosome XII intergenic site using pRSCG_ChXII | This study |
| *S. cerevisiae* M1744 + pRS42-G-DELTA + *T.r.eg2* | M1744-Δ-EG2 | *S. cerevisiae* M1744 with the *T.r.eg2* integrated at delta sites in the genome using pCas9NAT and pRS42G-DELTA | This study |
| *S. cerevisiae* M1744 + pRS42-G-DELTA + *T.e.cbh1* | M1744-Δ-CBH1 | *S. cerevisiae* M1744 with the *T.e.cbh1* integrated at delta sites in the genome using pCas9NAT and pRS42-G-DELTA | This study |
| *S. cerevisiae* MH1000 + pRS42-G-DELTA + *T.e.cbh1* | MH1000-Δ-EG2 | *S. cerevisiae* MH1000 with the *T.e.cbh1* integrated at delta sites in the genome using pCas9NAT and pRS42-G-DELTA | This study |
| *S. cerevisiae* MH1000 + pRS42-G-DELTA + *T.e.cbh1* | MH1000-Δ-CBH1 | *S. cerevisiae* MH1000 with the *T.e.cbh1* integrated at delta sites in the genome using pCas9NAT and pRS42-G-DELTA | This study |

**Table 2.** *Cont.*

| Strain | Abbreviation | Description | Reference |
|---|---|---|---|
| *S. cerevisiae* MH1000 + (*A.a.bgl1* + *T.r.eg2* + *T.e.cbh1*) | MH1000-B$_{11}$-E$_{10}$-C$_\Delta$ | *S. cerevisiae* MH1000 with *A.a.bgl1* targeted to Ch11, *T.r.eg2* targeted to Ch10 and *T.e.cbh1* targeted to the delta sequences | This study |
| *S. cerevisiae* MH1000 + (*A.a.bgl1* + *T.r.eg2* + *T.e.cbh1*) | MH1000-B$_{11}$-EC$_\Delta$ | *S. cerevisiae* MH1000 with *A.a.bgl1* targeted to Ch11, and, *T.r.eg2* and *T.e.cbh1* targeted to the delta sequences. | This study |

### 2.4. Yeast Transformation

All yeast transformations were carried out using the electroporation methods described by Cho et al. [23], with minor modifications to increase yeast cell permeabilization and thus improve transformation efficiency [24]. Briefly, harvested cells were washed with deionized distilled water, followed by resuspension in LiOAc/TE (0.1 M LiOAc, 10 mM TrisHCl pH 8.0, and 1 mM EDTA) solution. Resuspended cells were then incubated at 30°C for 45 min, prior to the addition of 20 µL 1 M DTT and further incubation with gentle shaking for 15 min at the same temperature. The mixture was then centrifuged, and cells were washed with deionized distilled water, followed by resuspension in electroporation buffer (1 M sorbitol, 20 mM HEPES). Competent cells were transformed with ~5 to 10 µg repair template DNA and 1 µg CRISPR plasmid DNA under standard conditions (1.4 kV, 200 ohms, 25 µF) using a micropulser (BioRad, Hercules, CA, USA). Following electroporation, cells were resuspended in 1 mL YPD broth media supplemented with 1 M sorbitol, followed by overnight incubation at 30 °C on an orbital shaker at 180 rpm. The transformation mixture was plated on YPDS solid media supplemented with CloNAT (100 µg/mL) and Geneticin (G418) (200 µg/mL) or Geneticin only as required for 2–3 days at 30 °C. For the 2-plasmid CRISPR-Cas9 system, *cas9*-carrying yeast strains were first created by transforming with the plasmid pCas9NAT (Table 1). These strains were subsequently transformed with the plasmid containing the specific target gRNA cassette and the relevant repair template. For the 1-plasmid system, *cas9*-free MH1000 and M1744 strains were transformed using the plasmids bearing both the *cas9* and gRNA cassettes as well as the relevant repair template (Table 1). Yeast strains were transformed with homology repair templates, the pCas9-NAT plasmid, and a CRISPR plasmid targeting a specific intergenic region on Chromosome X (pRS42-G_ChX), Chromosome XI (pRSCG_ChXI), or Chromosome XII (pRSCG_ChXII) as required for single gene integration (Table 1 and Table S2). The CRISPR plasmid pRS42-G-DELTA (targeting the genome-wide repeated "delta" sequences) was used to achieve multi-copy integration. The webtool E-CRISP was used to identify protospacer sites shown in Table S2 [25]. After sub-cultivation of transformants on selective plates, positive transformants were isolated and inoculated on YPD liquid media supplemented with CloNAT (100 µg/mL) and Geneticin (G418) (200 µg/mL) for strains transformed via the two-plasmid system and with only Geneticin (G418) (200 µg/mL) for strains transformed via the one-plasmid system for further screening.

These procedures were repeated for the purpose of introducing three genes (*T.r.eg2*, *T.e.cbh1* and *A.a.bgl1*) into a diploid yeast strain (MH1000) in successive rounds of transformation. The *T.r.eg2* gene was initially integrated into MH1000 using the above-mentioned procedure with pRS42-G-CHX or pRS42-G-DELTA. The confirmed transformants were streaked onto selective YPD media containing 100 µg/mL CloNAT to maintain the Cas9 plasmid in the yeast while eliminating the gRNA plasmid, followed by overnight incubation at 30 °C. The procedure of sub-cultivation was repeated for six days. After day 6, colonies from the original selective plates and the day 6 re-streaked plate were streaked on a new selective plate (containing G418) to confirm the loss of the G418 selective gRNA

plasmid in the sub-cultured strain. Following this curing of the gRNA plasmid and maintenance of the Cas9 plasmid, a second gene (*T.e.cbh1*) was transformed into this strain using the same method but with a different gRNA (pRS42-G-DELTA). After confirmation of transformation, the Cas9 and gRNA plasmids were cured by re-streaking the transformants on non-selective YPD media to eliminate both plasmids. We subsequently planned to target the *A.a.bgl1* to either the delta sequences or ChXI, but all further attempts to target the delta sequences failed. However, using this method, two strains containing a core set of cellulases were produced. MH1000-B$_{11}$-E$_{10}$-C$_\Delta$ contained *A.a.bgl1* targeted to ChXI, *T.r.eg2* targeted to ChX, and *T.e.cbh1* targeted to the delta sequences. MH1000-B$_{11}$-EC$_\Delta$ contained *A.a.bgl1* targeted to ChXI and *T.r.eg2*, as well as *T.e.cbh1*, targeted to the delta sequences.

*2.5. PCR Confirmation of Gene Integration and Positioning*

Hoffman and Winston's [26] total yeast DNA extraction method was used on randomly selected yeast transformants. Following that, PCR confirmation of the transformants was performed using the extracted DNA as a template. Taq DNA Polymerase Master Mix RED was used as directed by the manufacturer, using annealing temperatures indicated in Table S1. Different primers were used to confirm the presence of each transformed gene as well as the position of each gene in the genome. Table S1 contains detailed descriptions of the primers used to confirm the presence and position of each gene in the relevant genomic locus.

*2.6. Activity Screening and Quantitative Enzyme Assays*

*T.r.eg2* transformants were inoculated on YPD liquid media at 30 °C overnight. Cultures were then spotted on carboxymethylcellulose (CMC, Sigma Aldrich, St. Louis, MI, USA) solid media containing: 1% (*w/v*) carboxymethyl cellulose (CMC), 3 g/L Yeast nitrogen base with amino acids, 20 g/L glucose, 5 g/L ammonium sulphate and 20 g/L agar and incubated at 30 °C for 24 h. Following incubation, the plate was washed with water and stained with 0.1% Congo Red for 30 min, followed by washing with 1.2 M NaCl$_2$. For quantitative assays, transformed strains were cultivated in triplicate in 10 mL YPD media and cultivated at 30 °C for 48–72 h on an orbital shaker at 180 rpm for all assays. The samples were taken at 48- and 72-h cultivation times, and OD$_{600}$ readings were used to calculate the dry cell weight (DCW) for the strains [27]. Cells were removed by centrifugation, and the supernatant was used for the determination of the endoglucanase (EG) and cellobiohydrolase (CBH) activity, while the total culture was used for β-glucosidase (BGL) activity determination.

The dinitrosalicylic acid (DNS) EG assay was performed, as previously described by La Grange et al. [28], using 1% CMC as substrate. A standard curve was set using D-glucose at concentrations of 0.5–10 g/L. All spectrophotometric readings for the enzymatic assays were taken at 540 nm on a FLUOstar Omega Microplate Reader (BMG LABTECH, Ortenberg, Germany), and media blanks were included. The CBH activity of transformants was measured on the soluble fluorescent substrate 4-methylumbelliferyl-β-D-lactoside (MULac; Sigma) using the method previously described by Ilmén et al. [19] with a reaction time of 30 min at 37 °C and compared to a 4-methylumbelliferone (MU) standard curve set between 0.63 μM and 20 μM. The liberation of MU was detected by fluorescence measurement (excitation wavelength = 355 nm, emission wavelength = 460 nm). To evaluate the BGL activity of the recombinant strains, assays were carried out using *p*-nitrophenyl-β-D-glucopyranoside (pNPG; Sigma) as substrate, at 50 °C with a reaction time of 5 min as was previously described [29] and compared to a standard curve set between 0.075 and 1.25 mM pNP after centrifugation to remove cells. Assays were performed in biological and technical triplicates, and values were given as averages of these repeats with standard deviation indicated. For all enzyme assays, one unit was defined as the amount of enzyme required to produce one μmol of reducing sugar or equivalent in one minute under the assay conditions.

For the cellulose conversion assay, a substrate mixture containing 2% (*w/v*) Avicel PH101 (Sigma Aldrich), 50 mM sodium acetate (pH 5.0) and 0.02% (*w/v*) sodium azide

was made and continuously stirred to ensure homogeneity. In a 96-deep-well plate, a 1:1 mixture of the substrate mix and yeast culture supernatant was added to a final volume of 600 µL, with assays performed in triplicate. A sample was taken at 0 h as a baseline measurement for background sugar. The reaction was performed at 35 °C, shaking at 1000 rpm in a Heidolph Titramax 1000 microplate shaker/incubator (Heidolph, Schwabach, Germany) and sampled at 24 and 48 h. The amount of glucose in the supernatant was determined by performing a modified DNS assay method with absorbance determined at an OD of 540 nm, as previously described [29]. Using 72-h data, significant differences between activities attained were investigated using two-tailed *t*-tests, assuming unequal variance. A *p*-value lower than 0.05 was deemed significant.

*2.7. qPCR Gene Copy Number Analysis*

The integrated *T.r.eg2* and *T.e.cbh1* genes were quantified using real-time quantitative PCR (qPCR) by Inqaba Biotechnical Industries (Pty) Ltd. (Pretoria, South Africa). Primers specific for amplification of the target genes were used to determine the copy number of each cellulase expression cassette. The gene encoding $\alpha$-1,2-mannosyltransferase (*ALG9*) was selected to normalize the copy number of our genes of interest since it is present as a single copy in the haploid and as two copies in the diploid complement of the *S. cerevisiae* genome [30]. All DNA concentration measurements were carried out using the ND-1000 Spectrophotometer NanoDrop (Thermo-Fischer Scientific, Waltham, MA, USA). A standard curve was generated using a serial dilution of the pRDH180 (for *T.r.eg2* gene) and pMI529 for (*T.e.cbh1* gene) from 1 ng to 0.1 fg and with the parental strain for *ALG9* from 10 ng to 0.1 pg. qPCR was then performed in 96 well plates with Luna Universal qPCR Master Mix (New England Biolabs, Ipswich, MA, USA) using a dye-based qPCR assay. Each reaction contained 1 µL of DNA template, 0.25 µm forward and reverse primers and 1X Luna Universal qPCR Master mix. The reactions were run on a CFX96 Real-Time PCR System (Bio-Rad) following a standard two-step PCR program as suggested by the Luna Universal qPCR Master Mix manual. Three technical replicates were run for each DNA sample. The amplification of different input templates was evaluated based on the quantification cycle (Cq) value. Following that, the absolute copy number was calculated using a formula (Absolute copy number = DNA (g)/(g to bp const. $\times$ genome size). The average Cq values were plotted against the absolute copy number of standards, and standard curves were generated by linear regression of the plotted points. The absolute copy number for the strains was calculated based on the standard curves. The efficiency of the PCR was determined using the formula $E = -1 + 10^{(-1/\text{slope})}$, and the efficiency of all primer pairs used was over 95%. Standard melting curve analysis was performed to check the specificity of the qPCR products.

## 3. Results

### 3.1. Construction and Evaluation of Recombinants Strains Created via CRISPR-Cas9

To investigate the possibility of creating marker-free, enzyme ratio-optimized cellulose CBP strains of *S. cerevisiae*, we transformed haploid and diploid yeast strains using two different CRISPR systems. The first approach we tested was a two-plasmid system, in which *cas9* was expressed from a constitutive promoter on a low copy ARS4/CEN4-based vector, while the gRNA was expressed from an episomal 2 µ-based vector (Figure 1) [31]. A one-plasmid system was used in the second approach, in which *cas9* and the gRNA gene cassettes were both expressed from a multi-copy plasmid [32]. The second approach was chosen because of its versatility and recyclability, as well as its ability to reduce the number of transformation rounds and selectable marker usage. The CRISPR systems were designed to introduce double-strand breaks (DSBs) into specific intergenic sites on chromosomes X, XI or XII (Ch10, Ch11 or Ch12), allowing yeast to repair the breaks via homologous recombination. This was accomplished by using donor DNA that included cellulase reporter genes flanked by sequences homologous to either side of the relevant, targeted chromosome intergenic site (Figure 1C,D). Successfully transformed yeast strains

thus acquired plasmid-borne antibiotic resistance and a heterologous gene integrated into their genomes (Tables 1 and 2). To ensure that the correct transformants were obtained, all transformants were re-streaked on YPD selective plates, and random colonies were selected for further screening of the integrated reporter genes. Prior to PCR validation, *T.r.eg2* containing yeast transformants were first spotted on CMC agar plates to check for production of the heterologous EG indicated by clearing halo formation (Supplementary Figure S1A). The presence of the gene was subsequently confirmed via PCR analysis (Figure S1B). We also used PCR to confirm the presence of the *T.e.cbh1* gene in all CBH yeast transformants (Figure S1C). Finally, we PCR validated the presence of three genes (*T.r.eg2*, *T.e.cbh1* and *A.a.bgl1*) that were transformed into a single yeast strain to ensure that the reporter genes were present and stable (Figure S1D). All the colonies that showed appropriate bands in the agarose gel electrophoresis were then picked and cultivated in flasks, and their activities were subsequently evaluated through liquid assays.

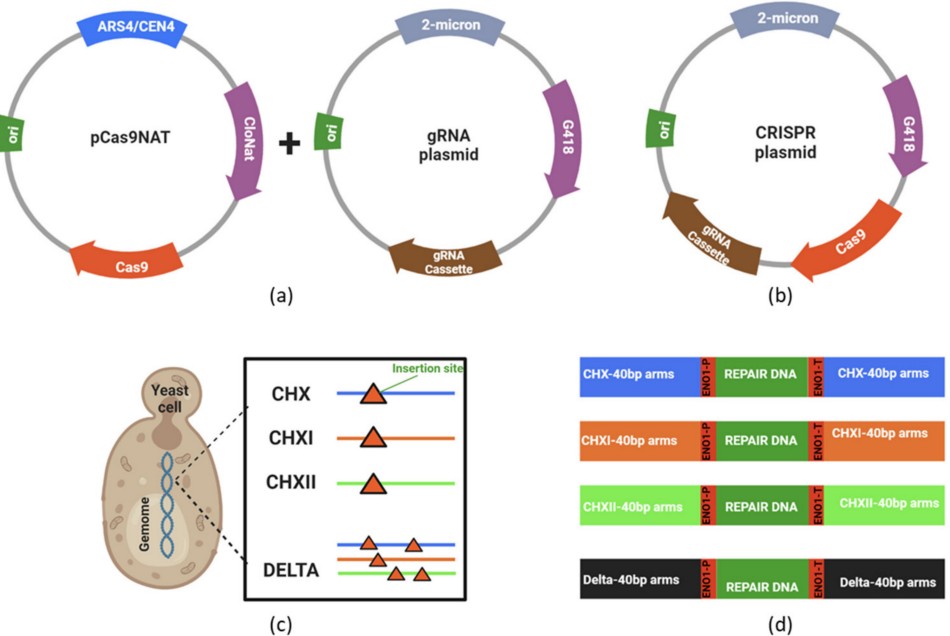

**Figure 1.** A schematic overview of the CRISPR-Cas9 systems used to transform a diploid industrial and a haploid laboratory *S. cerevisiae* strain. (**a**) Schematic illustration of the two-plasmid CRISPR system. A low-copy replicative (CEN/ARS-containing) plasmid contained the *cas9* gene and the selection marker for CloNAT resistance, and a multi-copy plasmid (2 μ) contained the gRNA expression cassette and the selection marker for G418 resistance. (**b**) Schematic illustration of the one-plasmid CRISPR system. A multi-copy (2 μ) plasmid contained the *cas9* gene, a gRNA cassette, and a selection marker for G418 resistance. (**c**) The CRISPR system targeted chromosomal intergenic sites for the integration of different genomic repair expression cassettes for gene editing. "Delta" represents the repeated Ty delta elements dispersed in the yeast genome that allows multi-target integration. Delta sequences are referred to by the symbol Δ elsewhere. (**d**) The repair DNA cassettes contained reporter genes flanked by the *ENO1* promoter, *ENO1* terminator, and 40-bp homology arms, targeting integration to the various genomic sites.

*3.2. Endoglucanase Integration and Activity*

To ascertain if different intergenic loci would lead to differences in heterologous protein production levels, we targeted single locus integration of *T.r.eg2* to different chromosomal sites in diploid and haploid *S. cerevisiae* strains. The intergenic chromosomal target sites selected were previously shown to support heterologous protein production at high levels [33]. After the *T.r.eg2* gene was confirmed to be present in the transformed MH1000 and M1744 strains via activity screening and PCR, strains were cultured to quantify their endoglucanase activities, as shown in Figure 2. The parental strain was included as negative

control and resulted in no activity, as expected. It was shown that both haploid and diploid *S. cerevisiae* strains had significantly lower activities when the gene was targeted to Ch10 or Ch11, compared to Ch12 ($p \leq 0.05$). At 72 h of cultivation, the highest activity for haploid M1744 strains was 160 U/gDCW for the Ch12 targeted strain, and the lowest activity was 40 U/gDCW for the Ch10 targeted strain. The highest activity observed in diploid MH1000 strains was 48 U/gDCW for the Ch12 targeted strain, and the lowest activity observed was 18 U/gDCW for the Ch11 targeted strain. Based on these findings, the haploid M1744 strain also had higher EG activities in all the targeted chromosome sites than the diploid MH1000 strain on a per DCW basis.

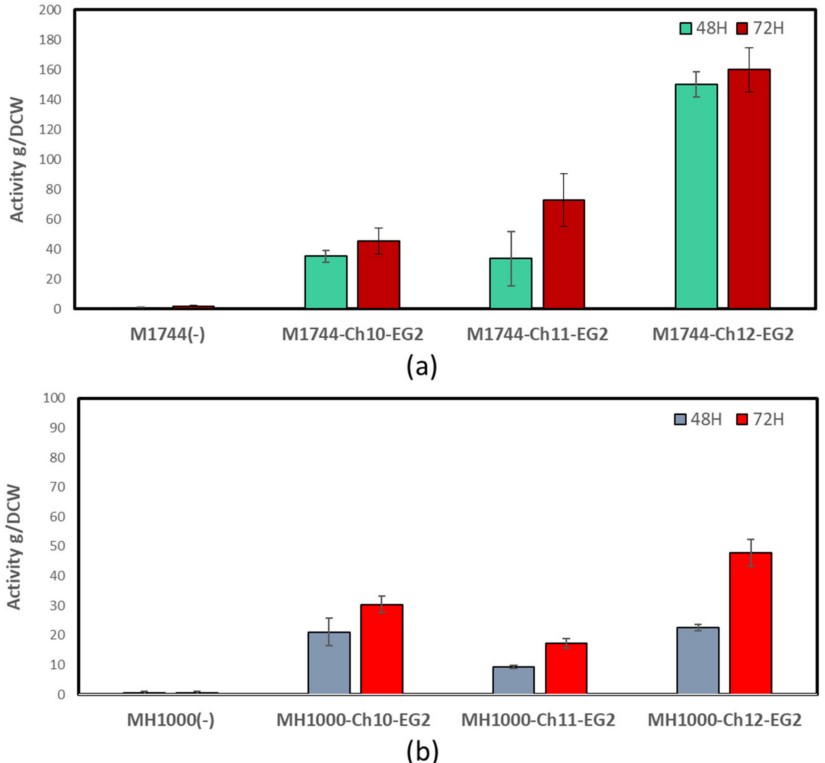

**Figure 2.** Endoglucanase activity profiles of recombinant yeast strains after 48 and 72 h cultivation. (**a**) The activity of EG2 producing haploid strains on CMC. (**b**) The activity of EG2 producing diploid strains on CMC. Values obtained were normalized with the dry cell weight of each specific yeast strain. The M1744 and MH1000 parental strains were used as negative control reference strains. The values given are the mean values of enzyme assays conducted in triplicate. Error bars indicate the standard deviation from the mean value for each strain.

Due to the differences in activity observed, we were interested in determining the number of *T.r.eg2* gene copies integrated into the genome of each transformant to disambiguate locus and copy number effects on expression level. This was determined using qPCR (Table 3). We expected the *S. cerevisiae* M1744 transformants to have one copy of the *T.r.eg2* gene because they are haploid with only one set of chromosomes and, thus, one target site. Since MH1000 *S. cerevisiae* strains are diploid and have two sets of chromosomes, we expected them to have at least two copies of the *T.r.eg2* gene. For targeting chromosomes X and XI, qPCR results matched our expectations as we confirmed one copy of *T.r.eg2* in the M1744 yeast strains and two copies in the MH1000 yeast strains. However, we were unable to accurately determine the copy number of *T.r.eg2* genes targeted to Ch12 from qPCR in MH1000-Ch12-EG2. Furthermore, the M1744-Ch12-EG2 strain was unexpectedly found to contain three copies of the gene. Evaluating the position of integration in these strains using standard PCR also indicated possible off-target integration. Our CRISPR systems achieved integration efficiencies of 55–65% for transformants expressing *T.r.eg2* at different sites in the yeast genome.

**Table 3.** Copy numbers of heterologous cellulase genes integrated at different chromosomal sites in the various transformants. Numbers after the strain names indicate different transformants of the same strain that were tested and match the numbers indicated in Figures 3 and 4.

| *T.r.eg2* Transformed Stains | *T.r.eg2* Copy Number |
| :---: | :---: |
| M1744-Ch10-EG2 (Haploid) | 1 |
| M1744-Ch11-EG2 | 1 |
| M1744-Ch12-EG2 | 3 |
| M1744-Δ-EG2-1 | 1 |
| M1744-Δ-EG2-2 | 1 |
| MH1000-Ch10-EG2 (Diploid) | 2 |
| MH1000-Ch11-EG2 | 2 |
| MH1000-Ch12-EG2 | N.D. |
| MH1000-Δ-EG2-1 | 3 |
| MH1000-Δ-EG2-1 | 1 |
| ***T.e.cbh1* transformed stains** | ***T.e.cbh1* copy number** |
| M1744-Ch10-CBH1 | 1 |
| M1744-Ch11-CBH1 | 1 |
| M1744-Ch12-CBH1 | 4 |
| M1744-Δ-CBH1-1 | 2 |
| M1744-Δ-CBH1-2 | 1 |
| M1744-Δ-CBH1-3 | 2 |
| M1744-Δ-CBH1-4 | 2 |

Subsequently, we targeted *T.r.eg2* to the delta-sequences for multi-copy integration of the gene in both haploid and diploid *S. cerevisiae* strains to potentially increase the integrated gene copy number. After confirmation of transformation, EG activities of the delta-integrated transformants were determined (Figure 3). As expression levels can vary significantly between different chromosomal regions [34], we tested eight positive transformants for each strain background to account for possible clonal variation due to the different positions in the genome where genes targeted to delta sequences may have integrated. Our CRISPR-delta-integration method was successful in integrating the *T.r.eg2* into *S. cerevisiae* haploid and diploid strains. Significant differences in activity between different delta-targeted transformants, as well as between delta-targeted transformants and single locus-targeted transformants, were observed, as confirmed by *t*-tests ($p \leq 0.05$). At 72 h of cultivation, the best performing haploid M1744-Δ-EG strain had an EG activity of 65 U/gDCW, while the best diploid MH1000-Δ-EG strain had an activity of 110 U/gDCW. The lowest EG activity observed in the haploid M1744 strains was 15 U/gDCW, while the lowest EG activity observed in the diploid MH1000 strains was 36 U/gDCW. Based on these findings, the diploid MH1000 strain outperformed the haploid M1744 strain in terms of EG activity levels when the gene was targeted to the delta sequences. It was also clear that significant clonal variation was evident in these transformants. Interestingly, single gene targeting to Ch10 or Ch11 in the haploid M1744 background led to similar and even higher EG activities compared to the delta-targeted strains. However, in the diploid strain background, EG activities of 3- to 5-fold higher were observed for some of the delta-targeted transformants, compared to their Ch10 or Ch11 targeted counterparts. We used qPCR to determine the number of *T.r.eg2* copies integrated into the delta sites (Table 3). M1744 strains that integrated *T.r.eg2* at the delta-sequences only contained 1 copy, while the highest copy number observed among MH1000 strains tested was 3.

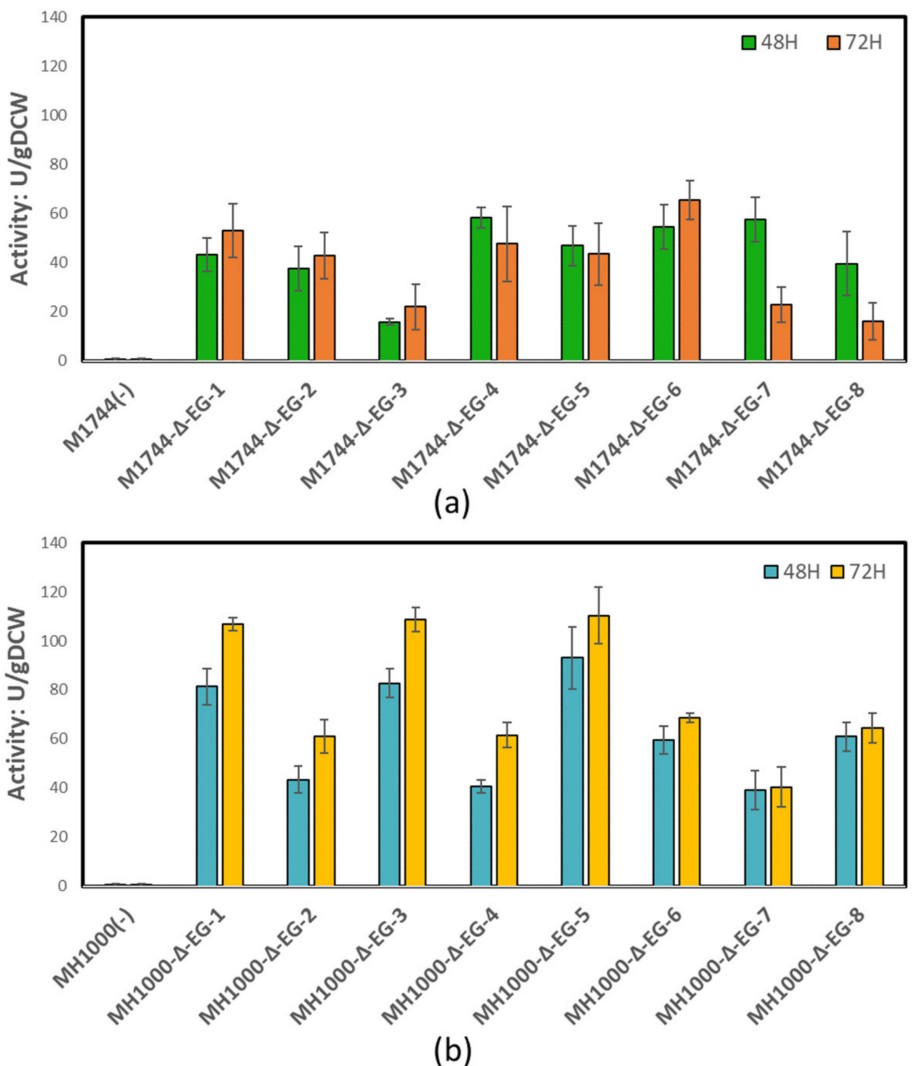

**Figure 3.** Endoglucanase activity profiles of recombinant yeast strains after 48 and 72 h cultivation. (**a**) The activity of EG2-producing haploid strains and (**b**) diploid strains on CMC. Numbers after the strain names refer to different transformants that were tested. Values obtained were normalized with the dry cell weight of each specific yeast strain. The values given are the mean values of enzyme assays conducted in triplicate. Error bars indicate the standard deviation from the mean value.

*3.3. Cellobiohydrolase Integration and Activity*

To evaluate the effect of the reporter protein on activities observed for integration at various loci, we investigated whether integration of a different cellulase gene, *T.e.cbh1*, would result in different outcomes. We repeated the integration experiments, targeting the same chromosomal sites in the M1744 haploid strain background. Transformants were confirmed via PCR (Figure S1) and cultivated on YPD to determine CBH activity. Delta integrated *T.e.cbh1* strains displayed significantly higher CBH1 activities than the single locus transformants where the gene was targeted to chromosome X,-XI, or-XII. At 72 h of cultivation, the highest CBH activity observed among the single locus targeted yeast strains was 110 mU/gDCW for the Ch12 targeted strain, while the lowest activity observed was 19 mU/gDCW for the Ch10 targeted strain. Our CRISPR system again yielded high integration efficiencies for targeting different sites of the yeast genome. In addition, all of the M1744 strains with *T.e.cbh1* targeted to the delta sites for multi-copy integration had higher activity than the single locus targeted M1744 CBH strains. The highest CBH activity observed at 72 h among the selected delta-integrated M1744 strains was 248 mU/gDCW, while the lowest was 210 mU/gDCW. Interestingly, there were no statisti-

cally significant differences between the single locus targeted transformants tested, but a significant difference between those strains and delta-targeted transformants was confirmed ($p \leq 0.05$). The activity of the delta-targeted transformants tested did not differ significantly.

We again determined the number of *T.e.cbh1* copies integrated into these transformants (Table 3). Varying integrated copy numbers were detected in these strains. Strains M1744-Ch10-CBH1 and M1744-Ch11-CBH1, targeted for single locus integration of *T.e.cbh1*, had one copy of the gene each, as expected, and displayed comparatively low CBH activity. However, the M1744-Ch12-CBH1 strain, which was also designed for single locus *T.e.cbh1* integration, contained 4 copies of the gene. As with the integration of the *T.r.eg2*, the Ch12 site presented a challenge for obtaining the desired integration, and we suspect off-target integration, as the integration locus for these transformants could not be confirmed via PCR, as opposed to those of the Ch10 and Ch11 targeted transformants.

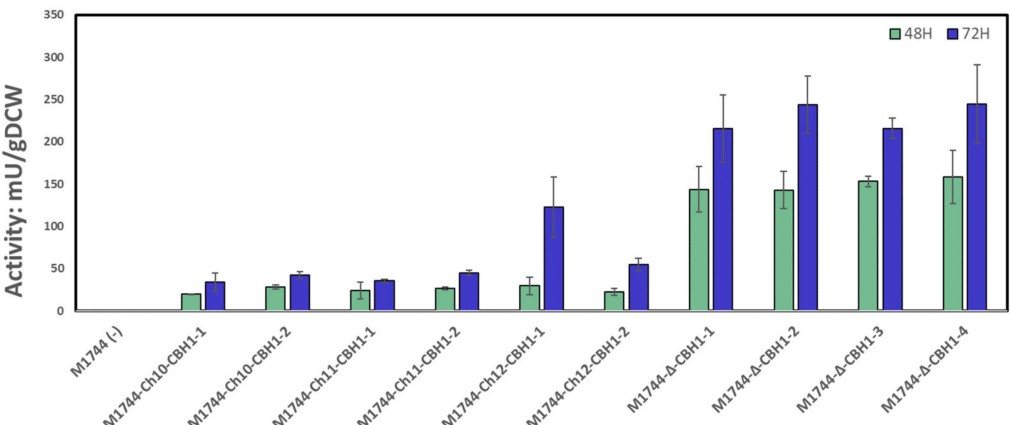

**Figure 4.** Cellobiohydrolase activity profile of CBH producing M1744 strains on MU-Lac after 48 and 72 h cultivations. Values obtained were normalized with the dry cell weight of each specific yeast strain. Numbers after the strain name refers to different transformants that were tested. The values given are the mean values of enzyme assays conducted in triplicates. Error bars indicate the standard deviation from the mean value for each strain.

### 3.4. Constructing Yeast Strains for Cellulose CBP

The transformation procedures were subsequently used to introduce three genes (*T.r.eg2*, *T.e.cbh1* and *A.a.bgl1*) into a diploid yeast strain (MH1000) in successive rounds of transformation. After confirmation of the transformants, we performed assays for the EG, CBH and BGL activities as well as an Avicel conversion assay for the 72-h cultivated MH1000-$B_{11}$-$E_{10}$-$C_\Delta$ and MH1000-$B_{11}$-$EC_\Delta$ strains (Figure 5a). For EG activity, the Ch10 targeted strain attained 51.5 U/gDCW, compared to 17.9 U/gDCW for the delta targeted strain. For CBHI activity (Mu-Lac), strain MH1000-$B_{11}$-$EC_\Delta$ displayed 0.5 mU/gDCW compared to 5 mU/gDCW for MH1000-$B_{11}$-$E_{10}$-$C_\Delta$ (Figure 5b). These values were significantly different. However, there was no significant difference in BGL activity between the two strains, with both strains attaining ~5 U/gDCW from their Ch11 targeted *A.a.bgl1* gene (Figure 5c). Both CBP strains were able to convert crystalline Avicel cellulose to glucose without the addition of exogenous enzymes (Figure 5d), but the MH1000-$B_{11}$-$E_{10}$-$C_\Delta$ was significantly more efficient. This strain attained an approximate 2.5-fold greater cellulose conversion compared to the MH1000-$B_{11}$-$EC_\Delta$ strain, likely due to its greater CBH activity.

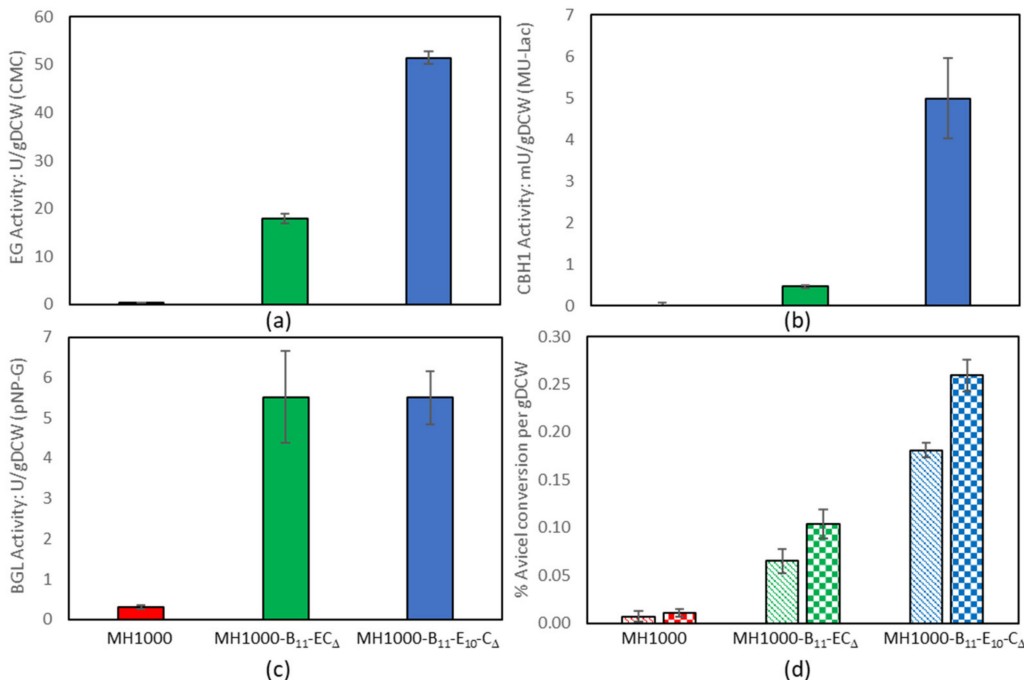

**Figure 5.** Enzyme activity profiles of recombinant yeast strains after 72 h cultivation. (**a**) The activity of EG2 producing diploid strains on CMC. The untransformed MH1000 strain was used as a negative control. (**b**) Enzyme activity profiles of CBH-producing strains on MU-Lac. (**c**) Enzyme activity profiles of BGL-producing strains on pNP-G. (**d**) Cellulose conversion of the strains measured on Avicel. In (**d**), bars with lines represent values measured after 24 h of the conversion assay, and bars with blocks represent 48-h values. All values obtained were normalized using the dry cell weight (DCW) of each strain at 72 h of incubation. All values represent mean values of assays conducted in triplicate with error bars indicating standard deviation.

## 4. Discussion

### 4.1. Endoglucanase Integration and Activity

Ronda et al. [17] previously reported that expression cassettes might integrate elsewhere in the genome via break-induced replication (BIR), resulting in strains where the gene of interest is difficult to localize. This is a known side-effect of some protospacer sequences in CRISPR applications. Despite using the E-CRISP predictive tool to avoid this problem, it still occurred for the Ch12 targeting site, though not for targeting the Ch10 or Ch11 sites. We observed, in agreement with previous reports, that the DSB created by the guide RNA-targeted Cas9 endonuclease was critical for correct integration at a significantly higher efficiency than endogenous homologous recombination alone [16].

As a diploid strain, MH1000 may be expected to offer more potential sites for *T.r.eg2* integration than the haploid M1744. The copy number data (Table 3) was thus in agreement with the observed higher levels of activity for the MH1000-Δ-EG2-1 strain compared to its single integration counterparts in MH1000 and the delta-targeted M1744 transformants (Figures 2 and 3). However, the different levels of activity of the single-copy transformants and the various delta-targeted transformants show that the locus of integration also had a significant effect on activity.

Numerous researchers have demonstrated that multi-copy integration via the delta sequences is an effective method for producing recombinant proteins, demonstrating increased expression levels using this method [35]. Although the number of integrated gene copies produced by delta-integration varies significantly according to its length, integration of 1 to 80 copies have been reported [36]. Sasaki et al. [37] integrated 40 copies of a 3.5 kb donor DNA sequence into the delta sequence using the CRISPR-Cas9 system of *S. thermophilus*. Shi et al. [38] also reported the breakdown of the delta sequence by the CRISPR system and subsequent delta integration. The highest copy numbers obtained

with the 8, 16, and 24 kb donor DNAs were 11, 10, and 18, respectively. In our study, we were only able to obtain 1–3 copies with our ~2 kb donor DNA. These differences may stem from differences in the strain backgrounds, differences in donor DNA length or the breakdown efficiency of our chosen delta-sequences by CRISPR-Cas9 [35] and/or chromosomal rearrangements, influenced by stressful environmental conditions [39].

### 4.2. Cellobiohydrolase Integration and Activity

As mentioned, it is likely that our Ch12 targeting yielded off-target integration. While this could be confirmed with whole genome sequencing, this was considered beyond the scope of our study. Jensen et al. [40] developed an efficient set of vectors, which enabled multiple simultaneous integrations of genes into specific "safe sites of insertion". The insertion sites are located between essential elements, which limits the occurrence of chromosomal aberrations due to the lethal effect this would cause [33]. We concluded that the Ch12 targeting sequence we selected might not be unique in the genome, allowing unwanted off-target DSBs in areas that lacked the presence of essential elements required for stable gene integration. Interestingly, the M1744-Ch12-CBH1 strain had double the amount of *T.e.cbh1* gene copies of the M1744-Δ-CBH1 strains, which exhibited significantly higher CBH activity. This again illustrated that the locus of integration might be of greater importance than simple copy number with regards to successful heterologous protein production, though the exact locus of this delta-targeted integration in the genome was not determined.

The two CRISPR-Cas9 systems used in this study were effective in producing recombinant proteins in both diploid and haploid strains of *S. cerevisiae*. However, we discovered that the two different genes (*T.e.cbh1* and *T.r.eg2*) we integrated into the yeast genome at different sites produced variable results in the M1744 strain background, as the sites with high EG activity were found to have lower CBH activity and *vice versa*. Furthermore, the delta-targeted sites performed better than all other sites at expressing high levels of CBH. However, in the EG transformants, the single locus targeted sites were more efficient at expressing EG than the CBH transformants that also had the gene integrated at the single locus sites. Wu et al. [41] discovered a correlation between the integration sites conferring the lowest and highest levels of expression. Low levels of expression were associated with the telomeres and centromeres, whereas high levels of expression have traditionally been associated with ARSs. In our study, intergenic chromosomal target sites were selected that were previously shown to support heterologous protein production at high levels [33]. However, it is clear that reporter protein-specific factors exist in obtaining high levels of expression of a specific gene of interest.

### 4.3. Constructing Yeast Strains for Cellulose CBP

Consolidated bioprocessing (CBP) is an improved process design to convert lignocellulosic feedstocks to biofuels in a single reactor using a single microbe or consortium [7,42]. Heterologous expression of genes encoding various types of cellulases from fungi has been reported in recombinant *S. cerevisiae* [7–10]. Constructing a CBP yeast strain requires the insertion of multiple genes into the genome of the host organism. This is often hampered by the availability of markers for subsequent rounds of engineering. Furthermore, for some applications, it is desired to have no DNA from bacterial origin remain in the industrial transformants. As CRISPR-Cas9 allows the insertion of a defined sequence without the need for leaving the selection marker in the transformant, it is an ideal technique for industrial strain development. This also allows the possibility of subsequent genetic modification in the same strain. We therefore set out to create diploid strains for CBP containing a core set of cellulase activities.

It was previously reported that optimal synergy between cellulases relies on different ratios of the enzymes [29,43]. The results shown in Figures 2–4 can thus inform the selection of loci for the integration of the cellulase genes. We observed that integrating *T.r.eg2* and *T.e.cbh1* to different loci yielded different levels of activity. These levels can thus be used to

help optimize the ratio of the cellulases produced. As CBHI activity is the most essential for crystalline cellulose hydrolysis and delta sequence targeted *T.e.cbh1* transformants yielded the highest CBHI activity, we targeted *T.e.cbh1* to that locus. It was previously shown that EG activity, though essential for synergism, was not required at very high levels [29]. We, therefore, targeted *T.r.eg2* to the Ch10 intergenic region but also created a version of the strains where this gene was targeted to the delta sequences for comparison. Finally, we attempted to target a BGL encoding gene (*A.a.bgl1*) to the delta sequences but were unsuccessful, possibly showing that the limit for integrating to that target in our strains was reached. The *A.a.bgl1* was thus targeted to the Ch11 intergenic region. In summary, this yielded two strains with a core set of cellulases, namely MH1000-B$_{11}$-E$_{10}$-C$_\Delta$ and MH1000-B$_{11}$-EC$_\Delta$ (Table 2). PCR amplification was used to confirm the presence of the four types of integrated genes in the CBP MH1000 yeast strains (Figure S1D).

The ten-fold difference in CBH activity between these two strains (Figure 5B), despite the gene being targeted to the same genomic locus, may be indicative of gene copy number differences, varying loci of integration, or that there was a limitation in the number of different genes that could be efficiently targeted to the delta sequences in our strains. However, we have demonstrated that an industrial *S. cerevisiae* strain could be successfully engineered with three cellulase-encoding genes without the need to maintain any selectable marker in the final strain. This makes the strain amenable to subsequent engineering steps. Furthermore, the observed differences in cellulase activities after integration at the different sites could be used to optimize the ratio of cellulases produced.

Multiplex genome editing is another option to be considered as we performed the introduction of the four different genes into a single yeast strain over the course of several transformation rounds. Creating suitable gRNAs and an appropriate vector to express them is critical in achieving multiplex integration and minimizing transformation rounds. [15,44]. Despite the significant advancements in the development of numerous assisting tools for designing gRNAs for various Cas proteins [45–48], the effectiveness of these gRNAs in vivo still has to be evaluated, which is even more critical for multiplex integration [49]. Additionally, the target locus selection has a substantial effect on the integration efficiency. In our study, we discovered that certain sites offered more activity for a certain gene than others. Baek et al. [50] discovered that specific target locations in gene-sparse regions were very ineffective due to restricted chromatin accessibility. Observations like these have been made elsewhere [49,51,52]. The identification and characterization of effective gRNAs and target loci will significantly increase the performance and efficacy of multiplex integration [49,50,53].

## 5. Conclusions

This study aimed to use CRISPR-Cas9 to integrate multiple genes at different genomic locations with the goal of creating markerless CBP-enabled yeast strains. Our CRISPR-Cas9 systems enabled efficient gene integration in different chromosomal positions within the yeast genome. However, we observed significant differences in activity at different loci: (i) when the same gene (*T.r.eg2*) was integrated and (ii) when a different gene was integrated (*T.e.cbh1*). Differences in gene copy number were also observed between haploid and diploid yeast strains. While we showed that targeting genes to our selected Ch12 site yielded transformants with higher levels of EG and CBH activity than all other single-copy integration sites, we were unable to confirm that the gene was integrated at the selected position, casting doubt over the usefulness of this particular locus for strain construction due to off-target integration. The repeated delta sequences were targeted for multi-copy integration, and we discovered that delta integration frequently resulted in transformants with only one gene copy in haploid strains and two copies in diploid strains for both genes tested. *T.e.cbh1* transformants where the gene was targeted to delta sites yielded higher activity than transformants that were targeted for single-copy integration. We also showed that while targeting *T.e.cbh1* to the delta sites yielded transformants with the highest CBH activity, *T.r.eg2* transformants had slightly lower EG activity when targeting

the delta sites in the haploid strain background. Our results clearly showed that there were gene-specific and locus-specific factors involved in obtaining high levels of heterologous protein production and that these factors cannot necessarily be predicted prior to testing the various loci and heterologous genes of interest.

Applying this knowledge, we created CBP-enabled yeast strains based on an industrial diploid yeast strain, transformed in different rounds of transformation to create a rudimentary cellulase system. Three cellulase-encoding genes could be transformed into the yeast strain in a marker-free strategy making additional genetic manipulation possible. Furthermore, the direct conversion of crystalline cellulose to monomeric sugars without the addition of exogenous enzymes could be shown. It was observed that targeting most of the genes to the delta sequences had detrimental effects on the individual cellulase activities and on cellulose conversion. It may therefore be prudent to identify genomic loci that allow high-level production of enzymes on interest and create transformants with combinations of optimally targeted genes rather than relying on delta integration alone. Based on these findings, we concluded that our system was useful and easily applied in the metabolic engineering of *S. cerevisiae* for biofuel production. It is likely also applicable to various wild-type and industrial strain isolates. While our rudimentary CBP strains were comparable to similar strains that were previously described (as reviewed in Den Haan et al., [12]; they did not degrade most of the available cellulose in the cellulose conversion assay. These strains could be further improved by using strain engineering approaches or applying the CRISPR-Cas9 approach to more process-amenable yeast strains [54,55]. The markerless nature of the CRISPR-Cas9 methodology enables a vast amount of additional engineering to improve these strains.

**Supplementary Materials:** The following supporting information can be downloaded at: https: //www.mdpi.com/article/10.3390/app122312317/s1: Table S1: Primers used in the study; Table S2: Gene integration target sites on different chromosomes; Figure S1: Confirmation of the cellulase genes integrated into haploid and diploid *S. cerevisiae* isolates.

**Author Contributions:** Conceptualization, R.d.H. and O.J.; methodology, R.d.H., G.R.v.L. and O.J.; formal analysis, R.d.H. and O.J.; writing—original draft preparation, O.J.; writing—review and editing, R.d.H. and G.R.v.L.; supervision, project administration, funding acquisition R.d.H.; All authors have read and agreed to the published version of the manuscript.

**Funding:** This research was funded by the National Research Foundation (South Africa) grant numbers 118894 and 92798.

**Acknowledgments:** We would like to acknowledge Rafeeqah Thompson and Philani Hadebe, whose Honours BSc projects (UWC) contributed to this study. We would like to thank W.H. van Zyl (Stellenbosch University) for providing yeast strains and plasmids. Finally, N. van Wyk is thanked for their help in procuring some of the CRISPR-Cas9 plasmids.

**Conflicts of Interest:** The authors declare no conflict of interest.

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
