# Peer review of "CRISPR-Based Multi-Gene Integration Strategies to Create Saccharomyces cerevisiae Strains for Consolidated Bioprocessing"

_applsci, doi:10.3390/app122312317_

Round 1
Reviewer 1 Report
The manuscript is well-written, and the data presentation is concise. I believe this article will be of interest to researchers in the field.
However, one of the significant issues with the manuscript is lack of any statistical analysis (e.g., Student’s t-test P-values etc.) to show that the differences observed are genuinely substantial. This deficiency severely affects the quality of the manuscript and needs to be addressed before acceptance for publication.
Author Response
We agree with the reviewer that adding statistical analysis will improve the validity of our results and conclusions. We have performed T-tests on all enzyme assay data. We have added sentences (lines 258 to 260) to describe the method. We have also discussed findings in the results in terms of their statistical significance i.e. p-values lower than 0.05 in (lines 338/9, 380-382, 418-422, 446-448)
Reviewer 2 Report
The manuscript by Jacob et al. describes the construction of haploid lab strains and diploid industrial strains of S. cerevisiae that express heterologous cellulase genes, by using the CRISPR-Cas9 system, as well as the evaluation of the ability to hydrolyze cellulose by a strain construct expressing three different cellulases. Engineered S. cerevisiae strains able to utilize cellulose have been obtained before, so this aspect of the manuscript is of limited novelty, and the existence of previous approaches and studies related to the final objective of the present study should be better emphasized in the Introduction section. The main relevance of the present study consists of using CRISPR technology for strains construction, which opens new possibilities for optimization of strains. The methodology employed and the results obtained are adequately described, and the information provided is of sufficient interest. Nevertheless, some aspects should be clarified.
-Fig. S1(d). The amplified DNA band for T.e. cbh1 in the constructed strain is very faint (lane 6), which deserves comment. Are the PCR conditions optimal for this amplification reaction?
-Table 3 and lines 347-349. There is a contradiction between the text and the figure with respect to the integration of the T.r. eg2 gene in the haploid strain Ch12. The table indicates that the copy number has been able to be determined (as 3), contrary to the text.
-Table 3 and Fig. 3. Strain names for delta-integrated constructs are different in the table and the figure, at least for the diploid integrates, which introduces confusion. Which is the correspondence between figure strains and table names? Where all integrates appearing in the figure tested for copy number, or only two of them (those showed in the table)?
-First paragraph of page 12. As I understand, what we should compare for single locus transformants are figures 2 and 4, not 3 and 4. If this is the case, it is not clear that for these single locus transformants an inverse relationship exists for eg2 and cbh1 integration in M1744.
-Table 3 and lines 411-416. The lower part of the table corresponds to cbh1 copies, not eg2 copies. Please unify strain nomenclature between Table 3 and Fig. 4.
-Section 3.4. While detailed description of the T.e. cbh1 gene integrates is made for the M1744 strain, sequential integration of the three genes is made in the diploid strain. Had integration of the cbh1 gene been also performed in the diploid strain (although not described in the manuscript)?
-Figure 4. Mainly in the case of CBH there is a large difference in enzyme activity between the two diploid strains tested. No data are presented on gene copy number or locus of integration, but this information could help to explain such differences.
Some minor points:
-Line 106. Delete ´2.2. Microbial cultivation´
-Line 266. Check the amounts of DNA template and primers in the qPCR Master mix
-Fig. 4. Specify also in the figure legends that the strains tested are M1744 derivatives
Author Response
Reviewer 2
The manuscript by Jacob et al. describes the construction of haploid lab strains and diploid industrial strains of S. cerevisiae that express heterologous cellulase genes, by using the CRISPR-Cas9 system, as well as the evaluation of the ability to hydrolyze cellulose by a strain construct expressing three different cellulases. Engineered S. cerevisiae strains able to utilize cellulose have been obtained before, so this aspect of the manuscript is of limited novelty, and the existence of previous approaches and studies related to the final objective of the present study should be better emphasized in the Introduction section. The main relevance of the present study consists of using CRISPR technology for strains construction, which opens new possibilities for optimization of strains. The methodology employed and the results obtained are adequately described, and the information provided is of sufficient interest. Nevertheless, some aspects should be clarified.
Authors’ response: We thank the reviewer for the thorough review of the manuscript and recognizing its value. In response to the statement that we should highlight previous approaches related to the objective in the introduction, we have added sentences between lines 70 and 78 to further summarize previous findings and motivate our study.
-Fig. S1(d). The amplified DNA band for T.e. cbh1 in the constructed strain is very faint (lane 6), which deserves comment. Are the PCR conditions optimal for this amplification reaction?
Authors’ response: The PCR conditions are likely not optimal for the T.e.cbh1 and A.a.bgl1 amplicons from genomic DNA, explaining their lighter bands on the gel compared to the plasmid based positive control in each case. However, as these bands are visible at the expected sizes and not present in the negative controls, we feel this is enough evidence of the presence of these heterologous genes in the transformants.
-Table 3 and lines 347-349. There is a contradiction between the text and the figure with respect to the integration of the T.r. eg2 gene in the haploid strain Ch12. The table indicates that the copy number has been able to be determined (as 3), contrary to the text.
Authors’ response: We meant for the lines pointed out by the reviewer to refer to the MH1000 based strain, but we neglected to mention the result of the M1744 haploid strain (which is the 3 copies mentioned above). To disambiguate this section, we have stated more pertinently what the results of each strain background were in lines 354 to 357.
-Table 3 and Fig. 3. Strain names for delta-integrated constructs are different in the table and the figure, at least for the diploid integrates, which introduces confusion. Which is the correspondence between figure strains and table names? Where all integrates appearing in the figure tested for copy number, or only two of them (those showed in the table)?
Authors’ response: We have addressed the inconsistencies between the strain names in the tables, figures, and throughout the text. Table 3 has been amended as well as the axis labels on figures 2, 3 and 4. All strain names were put in a “strain-target locus-gene” consensus e.g., M1744-Ch11-EG2, and updated in Table 2. These are now consistent throughout. As strain names in table 3 now match with those in Figures 2-4 direct comparisons are now easier to make. In several cases we tested more than one transformant to account for clonal variation (see lines 369-370); this is now made clear in the figure legends for figures 3 and 4 as well (lines 399 and 435).
-First paragraph of page 12. As I understand, what we should compare for single locus transformants are figures 2 and 4, not 3 and 4. If this is the case, it is not clear that for these single locus transformants an inverse relationship exists for eg2 and cbh1 integration in M1744.
Authors’ response: This should have referred to figures 2, 3 and 4. Initially we thought that the fact that we were seeing more EG activity in some of the single locus targeted strains compared to the delta-targeted strains (and the opposite for CBH1) in the haploid M1744 was significant. However, the observation that the Ch12 target led to off target integration and additional copies (Table 3) means that the statement can likely not be supported by the remaining data (without the Ch12 transformants and in light of T-test analysis for the other activities). We have therefore removed the “inverse relationship” statement on page 12.
-Table 3 and lines 411-416. The lower part of the table corresponds to cbh1 copies, not eg2 copies. Please unify strain nomenclature between Table 3 and Fig. 4.
Authors’ response: As mentioned above, we have addressed the inconsistencies between the strain names in the tables, figures, and throughout the text. Table 3 has been amended as pointed out by the reviewer, and all strain names were put in a “strain-target locus-gene” consensus e.g., M1744-Ch11-EG2.
-Section 3.4. While detailed description of the T.e. cbh1 gene integrates is made for the M1744 strain, sequential integration of the three genes is made in the diploid strain. Had integration of the cbh1 gene been also performed in the diploid strain (although not described in the manuscript)?
Authors’ response: The strains described in that section was made in an iterative fashion where one gene was integrated, CRISPR plasmid cured, transformation confirmed, and then the next gene integrated, etc. In this way, the eg2, cbh1 and bgl1 genes were put in, in that order. The procedure is described in detail in the methods section (page 6), starting in line 188.
-Figure 4. Mainly in the case of CBH there is a large difference in enzyme activity between the two diploid strains tested. No data are presented on gene copy number or locus of integration, but this information could help to explain such differences.
Authors’ response: Figure 4 refers to M1744 (haploid) transformants. If the reviewer meant figure 5 (5b) we agree that the copy number and locus of integration would influence the activities observed. We’ve now made this point clear in the discussion section (4.3; lines 552-555).
Some minor points:
-Line 106. Delete ´2.2. Microbial cultivation´
Authors’ response: corrected.
-Line 266. Check the amounts of DNA template and primers in the qPCR Master mix
Authors’ response: corrected.
-Fig. 4. Specify also in the figure legends that the strains tested are M1744 derivatives
Authors’ response: corrected.
Note: Changes in the document are indicated in red text. Several small editorial changes were also made. Figures have been improved for higher quality but no data on the figures were changes in any way.
-RdH

Round 2
Reviewer 1 Report
The authors have addressed the issues I had raised before. As such, I believe, the manuscript can be accepted for publication
Reviewer 2 Report
The authors have adequately answered the questions previously raised by the reviewer